# Study on the methodology of emergency decision-making for water transfer project contingencies: A case-based reasoning and regret theory approach

Feng Li[1], Xuewan Du[1], Xin Huang[2]*, Xiaoxia Fei[3]

1 School of Water Conservancy, North China University of Water Resources and Electric Power, Zhengzhou, China, 2 College of Agricultural Science and Engineering, Hohai University, Nanjing, China, 3 Henan Water & Power Engineering Consulting Co., Ltd., Zhengzhou, China

* yellowstar0106@163.com

## Abstract

To tackle the global water imbalance problem, a multitude of inter-basin water transfer projects have been built worldwide in recent decades. Nevertheless, given the complexity and safety challenges associated with project operation, effective emergency decision-making is crucial for addressing unforeseen incidents. Hence, this research has developed a two-stage emergency decision-making framework to tackle the uncertainty in the development trends of emergencies in inter-basin water transfer projects. (1) The first stage mainly utilizes case-based reasoning techniques to extract historical case information and disposal plans for inter-basin water transfer projects. Subsequently, a holistic similarity model is built by employing structural similarity and local attribute similarity algorithms to identify highly similar historical cases. (2) The second stage involves the optimization and adjustment of decision-making plans based on the dynamic evolution characteristics of emergencies. It utilizes the theory of decision-makers regret psychology and combines it with practical case studies to verify the scientific rationality of the method. This enables it to achieve effective multidimensional expression and rapid matching of scenarios, satisfying the decision-making requirements of "scenario response". Finally, this study compares the results obtained from this method with those computed using the traditional TOPSIS method and fuzzy comprehensive evaluation method, further validating its feasibility and effectiveness. In practice, this method can provide effective support for decision-makers work.

## Introduction

Cross-basin water transfer projects refer to major water conservancy projects that cross two or more river basins or regions to carry out significant water diversion or transfer [1], and will significantly affect water supplies, hydrology, and the environment in both donor and receiving basins [2–4]. In recent years, a number of major inter-basin and inter-regional water diversion and transfer projects have been built around the world in order to promote water disaster

**Data Availability Statement:** All relevant data are within the manuscript (Table 1).

**Funding:** The author(s) received no specific funding for this work.

**Competing interests:** The authors have declared that no competing interests exist.

prevention and control, water resource conservation, water ecology protection and restoration, and water environment management, as well as to solve the problems of global summer floods and winter droughts, the lack of the north and the south, and the imbalance in the distribution of water resources in time and space. It provides strong support for addressing insufficient water resource demand, and water scarcity [5], and promoting high-quality development of water conservancy in a new stage, thus providing strong support for sustained and healthy economic and social development [6, 7].

According to statistics, there are currently over 160 water transfer projects worldwide [8, 9], all of which have played an important role in flood control, alleviating water shortages in various regions. As a major water transfer project in China, the South-to-North Water Diversion Project has transferred 65.4 billion cubic meters of water since the completion of the main construction of the Middle Route Phase I and the Eastern Route, directly benefiting as many as 176 million people and replenished 10 billion cubic meters of water for ecological purposes [10]. It has played an important role in economic and social development as well as ecological environment protection, serving as a strategic infrastructure project fundamentally addressing water scarcity in North China and Northwest China [9]. However, the inter-basin water transfer project is a typical interconnected system engineering. The route of middle route of the South-to-North Water Diversion Project shown in Fig 1 has the characteristics of crossing different river basins and a long water diversion route. Once construction safety and quality issues or sudden events such as water resource pollution occur, it is highly likely to cause huge environmental and economic losses to the project itself or the surrounding environment [11, 12], often accompanied by significant hidden dangers, and may even result in serious injuries or deaths. Therefore, strengthening research on emergency rescue and disposal decision-making for major water diversion projects, establishing effective emergency management measures for sudden events, ensuring the normal and continuous operation of the project, maintaining the safety of people's lives and property, and promoting the stable development of the social economy are of great significance and practical importance.

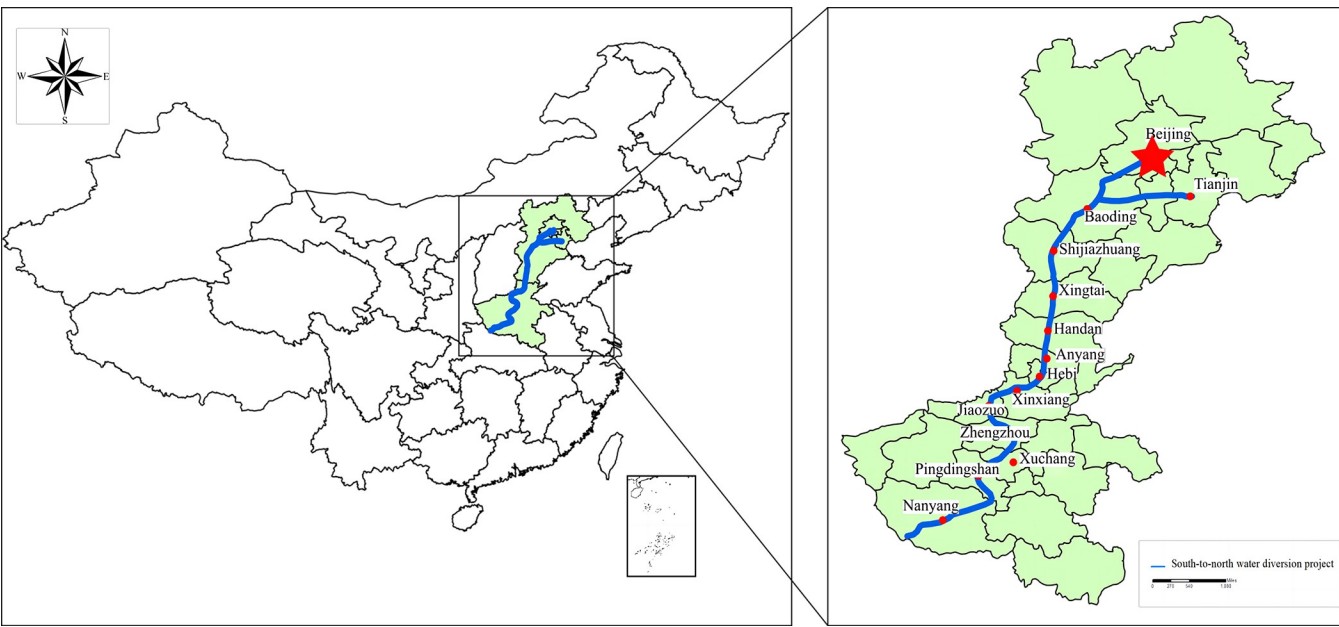

**Fig 1. Schematic diagram of the middle route of south-to-north water diversion project.**

The serious consequences of environmental emergencies continually remind governments of the importance of emergency decision-making, rousing meaningful exploration and practice around the emergency organization structure, legal system, and other aspects [13–15]. Currently, there are a large number of Inter-basin water transfer projects that have been the subject of numerous studies [16–19]. However, emergency response and decision-making, which are essential components of emergency management, still lack sufficient support in the research on emergency response to sudden incidents in cross-basin water transfer projects. When a disaster occurs, decision-makers must develop emergency action plans within a limited time to maximize the objectives of emergency decision-making. However, due to the complex and intertwined causes of sudden events, decision-makers not only need to identify and anticipate risks before an event occurs but also need to make timely and effective solutions when a danger arises. Under such immense pressure, the dynamic disposal process based on emergency decision-making [20] is likely to lead to failed emergency measures that are difficult to handle using conventional methods. Therefore, it is crucial to improve the quality and efficiency of emergency decision-making by establishing emergency decision-making models that are suitable for the characteristics of sudden events. Consequently, case-based reasoning technology has emerged and matured, being applied in multiple fields [13, 21–23]. Additionally, in existing research, scholars have mostly focused on selecting emergency plans based on the expected utility of one-time decision plans, without considering the expected regrets of the plans. In practical decision-making processes, decision-makers not only consider the expected utility of emergency plans but also take into account the expected regrets of overreacting. Therefore, in case-based decision-making, decision-makers often exhibit the following psychological behavior: when retrieving the most similar historical cases, if they find that choosing other historical cases is more similar to the target case, they will feel delighted with their decision. Conversely, they will feel regretful. Compared to current commonly used methods such as prospect theory [24, 25], regret theory does not require decision-makers to provide reference points and involves fewer parameters in calculations, making it computationally simpler [26]. In recent years, this theory has been widely applied in fields such as group decision-making and hesitant fuzzy multi-criteria decision-making [27], effectively avoiding decision-makers selection of historical cases that would cause regret [28].

In conclusion, this study focuses on the uncertainty and multiple attribute issues of emergency events in inter-basin water transfer projects. Taking into consideration the decision-makers regret psychology, a case-based reasoning approach is adopted to analyze the decision-making of emergency events in a specific section of the South-to-North Water Diversion Project based on historical experience, data, and knowledge inference. This study makes three main contributions: (1) In response to the uncertainty of the development trends of emergency events in inter-basin water transfer projects, a two-stage emergency decision-making plan is constructed. (2) By fully considering decision-makers regret avoidance behavior, the theoretical and practical significance of regret theory in selecting emergency response plans for sudden events is studied and compared with other methods to verify its applicability. (3) Strict differentiation is applied to accident information indicators in emergency events of inter-basin water transfer projects, and hesitant fuzzy numbers are used to represent decision-making information, which enhances the effectiveness of emergency response.

The structure of the remaining parts of this study is as follows. Part 2 introduces the research methodology of this paper and proposes research questions and solutions based on it. Part 3 takes the decision-making research on the siphon backflow emergency event in a certain section of the South-to-North Water Diversion Project as an example to validate the applicability of the research methodology. Part 4 selects the TOPSIS method and fuzzy comprehensive evaluation method to optimize the selection of preliminary plans and compares them with

regret theory. Finally, Part 5 draws conclusions based on the entire paper, while summarizing the main contributions and shortcomings as well as suggesting directions for further research.

## Research methodology

Case-Based Reasoning (CBR) theory is a method that solves current case problems by using the solutions to similar historical cases. It helps decision-makers make choices based on similarity and implementation efficiency. If the historical cases used for reference are less relevant to the target case, the generated solutions based on the historical cases will also be less effective. Therefore, applying solutions from similar historical cases with poor implementation effects to the target case could result in poor decision-making [29]. Through case reasoning, decision-makers reasoning speed can be improved, the efficiency of emergency rescue and disposal decision-making can be increased, and the feasibility of the final plan can be ensured. To solve the problem of emergency decision-making for sudden events in the South-to-North Water Diversion Project, this study describes and analyzes the problem using this method. In the following sections, some related background knowledge is introduced to make the method more accessible.

### Problem description

In the event of an unexpected incident, decision-makers retrieve historical cases from the case repository, initially assuming that the emergency case repository for a cross-basin water transfer project is denoted as $Z = \{Z_1, Z_2,\ldots, Z_{i,\ldots}, Z_m\}$, where $A_i$ represents the $i$-th case. Each emergency case corresponds to characteristic information $C = \{C_1, C_2, \cdots, C_j, \cdots, C_n\}$ of the disaster for the sudden water transfer project event, where $C_j$ represents the $j$-th disaster characteristic information. The weight of each disaster information attribute is denoted as $W^k = \{w_j^k | j = 1, 2, \cdots, n\}$, where $w_j^k$ represents the weight of attribute $C_j$ for the $j$-th property, satisfying $0 \leq w_j^k \leq 1$ and $\sum_{j=1}^{n} w_j^k = 1$. Adopting $q_i = \{q_{ij} | j = 1, 2, \cdots, n\}$ to denote the vector of problematic attribute values about the historical case $Z_i$, with $q_{ij}$ representing the attribute value of historical case $Z_i$ concerning the feature information indicator $C_j$. Adopting $q_0 = \{q_{0j} | j = 1, 2, \cdots, n\}$ to denote the problem attribute value vector related to the target case $Z_0$, with $q_{0j}$ representing the attribute value of target case $Z_0$ concerning the feature information indicator $C_j$. According to the current operation status of the water diversion project, combined with weather forecast information and the surrounding environmental conditions, it is inferred that unexpected events may occur in this section of the water diversion project. Subsequently, based on the emergency process of unexpected events and in conjunction with emergency practices for water diversion projects, it is necessary to initiate an initial emergency response plan and make necessary preparations. Based on this, this article mainly addresses the following two issues:

**(1) Question 1:** How to generate an initial emergency plan from the emergency plan repository based on the current operation status of the water diversion project?

Once the emergency plan is generated, emergency response is necessary. After exchanges and discussions between emergency decision-making experts, rescue commanders, and on-site management leaders, the future development scenarios $S = \{S_1, S_2, \cdots, S_l\}$ of the sudden event are predicted. Where $S$ represents a collection of $l$ possible scenarios that may occur in future water transfer project accidents. Where $S_t$ represents the $t$-th scenario that may appear after the occurrence of the accident. Where $p_t$ denotes the probability of scenario $S_t$, satisfying

$p_t \in [0,1]$, $\sum_{t=1}^{l} p_t = 1$ and $t \in l$. This study takes into account the potential impact of

implementing different initial disposal plans on the development of accident scenarios, and thus, different implementation plans may result in varying evolutions of accident scenarios and associated losses.

In anticipation of future accident scenarios, let us assume that the emergency decision rescue team has developed $m$ emergency response plans. Where $A_i$ represents the $i$-th emergency response plan ($i\in M$). In the course of emergency rescue operations, the assessment of emergency response plans must take into account the evaluation criteria encompassing personnel casualties, economic losses, and social impacts. Notably, personnel casualties and economic losses are categorized as cost-type attributes, whereas social impact is considered a benefit-type attribute, denoted as $D = \{D_1, D_2, \cdots, D_n\}$, with $D_j$ denoting the $j$-th attribute ($j\in N$).

The level of satisfaction of attribute $D_j$ after the implementation of emergency rescue plan $A_i$ under scenario $S_t$ is represented by $\tilde{h}_{ij}^t$. Due to the presence of numerous uncertainty factors following water transfer project accidents and the lack of consensus among experts, $\tilde{h}_{ij}^t$ is treated as a hesitant fuzzy number. Specifically $\tilde{h}_{ij}^t = H\{\gamma_{ij}^{1(t)}, \gamma_{ij}^{2(t)}, \cdots, \gamma_{ij}^{l(t)}\} = H\{\gamma_{ij}^{\lambda}|\lambda = 1, 2, \cdots, l_{ij}(t)\}$, where $l_{ij}(t)$ denotes the number of elements in $\tilde{h}_{ij}^t$, representing the count of decision-making expert opinions ($\gamma_{ij}^{\lambda} \in [0, 1]$). By processing all decision information, a hesitant fuzzy decision matrix $H = [\tilde{h}_{ij}^t]_{m\times n}$ ($i\in M, j\in N, t\in l$) is constructed.

**(2) Question 2:** How can decision-makers dynamically adjust in a risk environment based on the dynamics of water diversion project accidents and the uncertainty of information? In the following chapters, this study specifically addresses the above issues.

## Background knowledge

In order to address the above-mentioned issues, this article uses a research methodology that combines Case-Based Reasoning (CBR) and regret theory to seek solutions to decision problems in water diversion project emergencies. Therefore, in this section, the concepts and definitions of scoring function, deviation function, utility function, and regret theory are introduced in sequence, as well as their meanings in the research on emergency decision-making problems, in order to better understand the research methodology of this article.

**Scoring function.**  The scoring function quantifies and scores the features of emergency decision-making solutions based on specific evaluation indicators or criteria, aiming to provide a quantitative assessment tool for emergency decision-making, helping decision-makers compare and select different decision-making solutions. In recent years, the scoring function has been recognized as an important tool for studying interval intuitionistic fuzzy multi-criteria decision-making problems, attracting significant attention from numerous experts and scholars and yielding many research achievements. According to Wang Xuefang, a reasonable and effective scoring function needs to consider membership, non-membership, hesitation, and their tendencies simultaneously [30]. Based on this, the definition of the scoring function used in this paper is as follows:

**Definition 1:** Let $X$ be a given domain. Let $H = \{< x, \tilde{h}_H(x) > |x \in X\}$ be the hesitant fuzzy set on $X$. Where $\tilde{h}_H(x)$ is a set consisting of several real numbers in the closed interval [0,1], representing the membership degrees of $x\in X$ in the set $H$. Call $\tilde{h} = \tilde{h}_H(x)$ a hesitant fuzzy number, the detailed formula is $\tilde{h} = \tilde{h}_H(x) = H\{\gamma|\gamma \in \tilde{h}_H(x)\} = H\{\gamma^1, \gamma^2, \cdots, \gamma^{l\tilde{h}}\} = H\{\gamma^{\lambda}|\lambda = 1, 2, \cdots, l\tilde{h}\}$, where $\gamma^{\lambda} \in [0, 1]$, and $l\tilde{h}$ represent the number of elements in $\tilde{h}$. Unless otherwise specified, the elements in $\tilde{h}_H(x)$ are arranged in ascending order, and $\tilde{H}$ denotes the collection of hesitant fuzzy numbers.

**Definition 2:** Let $\tilde{h} \in \tilde{H}$ be a function, and its scoring function $s(\tilde{h})$ is defined as:

$$s(\tilde{h}) = \frac{1}{l\tilde{h}} \sum_{\gamma \in \tilde{h}} \gamma \tag{1}$$

In the equation, $l\tilde{h}$ represents the cardinality of set $\tilde{h}$.

When given $\tilde{h}_1, \tilde{h}_2 \in \tilde{H}$, if $s(\tilde{h}_1) > s(\tilde{h}_2)$, then $\tilde{h}_1 \succ_s \tilde{h}_2$. if $s(\tilde{h}_1) < s(\tilde{h}_2)$, then $\tilde{h}_1 \prec_s \tilde{h}_2$. if $s(\tilde{h}_1) = s(\tilde{h}_2)$, then $\tilde{h}_1 \sim_s \tilde{h}_2$. It should be noted that in many cases, the ranking of $\tilde{h}$ based on the scoring function $s(\tilde{h})$ is ineffective, $s(\tilde{h})$ only takes into account the average level of elements in $\tilde{h}$, which may result in the loss of certain decision information.

**Bias function.** The deviation function is a function used to describe the cognitive biases or decision biases that decision-makers may have in emergency decision-making. It measures the difference between the decision-maker's decision and the rational decision, considering this difference as a manifestation of decision bias. The application of the deviation function in emergency decision-making research can help decision-makers identify and diagnose decision biases, and then propose corresponding decision support methods and intervention measures. In specific decision analysis models, the deviation function is usually defined together with utility functions or loss functions, such as prospect theory, mental accounting, and regret theory, helping decision-makers integrate considerations of benefits and risks to make optimal emergency decisions. Here is the definition of the deviation function:

**Definition 3:** Given $\tilde{h} \in \tilde{H}$, $\tilde{h} = \tilde{h}_H(x) = H\{\gamma | \gamma \in \tilde{h}_H(x)\} = H\{\gamma^\lambda | \lambda = 1, 2, \cdots, l\tilde{h}\}$, the deviation function $v(\tilde{h})$ is defined as follows:

$$v(\tilde{h}) = \frac{1}{l\tilde{h}} \sum_{\lambda=1}^{l\tilde{h}} |\gamma^\lambda - s(\tilde{h})| \tag{2}$$

In the equation: $l\tilde{h}$ represents the number of elements in $\tilde{h}$. $\gamma^\lambda$ is the $\lambda$-th smallest element in $\tilde{h}$.

In this context, $s(\tilde{h})$ and $v(\tilde{h})$ correspond to the mean and variance in statistics. $s(\tilde{h})$ reflects the average level of all elements in $\tilde{h}$, while $v(\tilde{h})$ reflects the degree of deviation of each element in $\tilde{h}$ from the mean value, which indicates the level of disagreement in decision opinions.

**Utility function.** The utility function is a function used to evaluate the relative value of different decision options. It maps the outcomes of emergency decisions onto a numerical range, reflecting the utility or benefits to decision-makers from different strategies. In the dynamic adjustment of emergency plans, different decision-makers have varying utilities for adjusting costs, adjustment losses, and disposal effects of emergency plans, meaning that the corresponding utility functions are not the same. Specifically, the utility function maps the decision outcomes to a real number domain, reflecting the value or utility of the results for decision-makers. Since the utility function embodies the preferences of decision-makers, utilizing it to perform utility analysis on various adjustment plans leads to decision outcomes that are more in line with reality [31]. The utility function used in this paper is defined as follows:

**Definition 4:** Let $\tilde{h} \in \tilde{H}$, $\tilde{h} = \tilde{h}_H(x) = H\{\gamma | \gamma \in \tilde{h}_H(x)\} = H\{\gamma^\lambda | \lambda = 1, 2, \cdots, l\tilde{h}\}$. Its deviation function is $v(\tilde{h})$, and its utility function $u(\tilde{h})$ is defined as:

$$u(\tilde{h}) = \left( \frac{s(\tilde{h})}{1 + v(\tilde{h})} \right)^\alpha \tag{3}$$

In the equation, parameter $\alpha(0<\alpha\leq1)$ is a constant given by the decision-maker according to practical requirements. It is clear that $u(\tilde{h})$ is a monotonically increasing concave function.

It should be noted that the utility function $u(\tilde{h})$ uses the scoring function $s(\tilde{h})$, deviation function $v(\tilde{h})$, and parameter $\alpha$ to jointly measure the magnitude of hesitant fuzzy decision information. $s(\tilde{h})$ reflects the overall level of $\tilde{h}$, with larger $s(\tilde{h})$ resulting in larger $u(\tilde{h})$. $v(\tilde{h})$ reflects the degree of divergence of decision information, with smaller $v(\tilde{h})$ indicating smaller degree of divergence in decision opinions, resulting in larger $u(\tilde{h})$. $\alpha$ is a parameter given in advance by the decision-maker based on decision needs. Particularly, when $\alpha = 1$, $u(\tilde{h})$ degenerates into the index of group satisfaction.

Here, $u(\tilde{h})$ has the following properties:

1. In the case of $\tilde{h} \in \tilde{H}$, $\tilde{h} = \tilde{h}_H(x) = H\{\gamma | \gamma \in \tilde{h}_H(x)\} = H\{\gamma^\lambda | \lambda = 1, 2, \cdots, l\tilde{h}\}$, and $u(\tilde{h}) \in [0, 1]$ is applicable.

2. For a single-valued hesitant fuzzy number, denoted as $\tilde{h} = \{\gamma\}$, the utility is $u(\tilde{h}) = (\gamma)^\alpha$. For an empty hesitant fuzzy number, denoted as $\tilde{h} = \{0\}$, the utility is $u(\tilde{h}) = 0$. For a full hesitant fuzzy number, denoted as $\tilde{h} = \{1\}$, the utility is $u(\tilde{h}) = 1$.

Proof:

1. $0 \leq u(\tilde{h}) = (\frac{s(\tilde{h})}{1+v(\tilde{h})})^\alpha \leq (s(\tilde{h}))^\alpha \leq 1$.

2. If $\tilde{h} = \gamma$, that is $s(\tilde{h}) = \gamma$, $v(\tilde{h}) = 0$, then $u(\tilde{h}) = (\gamma)^\alpha$. If $\tilde{h} = \{0\}$, that is $s(\tilde{h}) = 0$, $v(\tilde{h}) = 0$, then $u(\tilde{h}) = 0$. If $\tilde{h} = \{1\}$, that is $s(\tilde{h}) = 1$, $v(\tilde{h}) = 0$, then $u(\tilde{h}) = 1$.

**Regret theory.** In the regret theory of emergency decision-making, it is believed that decision-makers make decisions to minimize potential regret in the future. Graham and Robert proposed the fundamental model of regret theory in 1982 [32], which introduces "regret" based on expected utility theory. It takes the individual's psychological experience of outcomes as a baseline when "doing nothing" and incorporates a regret function, leading to a modified model of expected utility function. Let $x$ and $y$ represent the outcomes obtained from choosing options $A$ and $B$, respectively. The decision-maker's perceived effect of option $A$ is:

$$U(x, y) = V(x) + R(V(x) - V(y)) \tag{4}$$

Among them, $V(x)$ and $V(y)$ respectively represent the utilities that the decision-maker can obtain from the outcomes of options $A$ and $B$, and $R(V(x)-V(y))$ is the regret-rejoicing value. When $R(V(x)-V(y))>0$ is a rejoice value, it indicates that the decision-maker feels rejoice for choosing option $A$ over option $B$. When $R(V(x)-V(y))<0$ is a regret value, it indicates that the decision-maker feels regret for choosing option $A$ over option $B$. Here, the regret-rejoicing function is a monotonically concave function, and when $V(x)-V(y) = 0$, then $R(V(x)-V(y)) = 0$.

This modified utility function incorporates the factor of regret, which can better explain decision phenomena that deviate from the traditional expected utility function theory as found in many empirical studies. Later, Quiggin further extended this theory to the case of a general choice set, enabling it to include the selection of the optimal option from a set of multiple decision alternatives.

Let $A_1, A_2, \cdots, A_m$ be a set of $m$ alternative solutions, where $A_i$ represents the $i$-th alternative solution, with $i = 1, 2, \cdots, m$. Let $x_1, x_2, \cdots, x_m$ represents the results of solution $A_1, A_2, \cdots, A_m$,

where $x_i$ represents the result of solution $A_i$. Then, the decision maker's perceived utility for solution $A_i$ is:

$$U_i = V(x_i) + R(V(x_i) - V(x^*)) \qquad (5)$$

Where $x^* = \max\{x_i\}$, $R(V(x_i)-V(x^*)) \geq 0$ indicates a rejoicing value and $R(V(x_i)-V(x^*)) \leq 0$ represents a regretful value. The regret-rejoicing function still follows a monotonically increasing concave pattern.

## Model construction

In this chapter, from a two-stage perspective, we study how decision makers should generate preliminary decision plans and how to make dynamic adjustments. The first stage is based on CBR theory to generate preliminary emergency decision plans. The second stage is based on regret theory, combined with hesitant fuzzy utility function, to calculate the regret value of each emergency plan. Then, the comprehensive utility value and regret value are integrated to obtain the perceived utility of each emergency plan. Finally, the plans are ranked using perceived utility and emergency plans are adjusted according to the actual situation. The specific process is shown in Fig 2.

**Stage 1:** Existing cross-basin water transfer project accident cases are integrated as a case library, and the indicator information of historical cases is extracted to describe the current information based on weather forecast information and the status of historical accident cases. In addition, due to the complexity and diversity of cases, their attribute values may exist in various forms. Therefore, this paper considers using a heterogeneous calculation method that combines qualitative and quantitative aspects as the attribute type for the problem. This method includes four types: symbolic, ordinal enumeration, precise numerical, and fuzzy linguistic types. For instance, in the case of accidents in cross-basin water transfer projects, the evaluation value of "whether there is flood discharge upstream" is calculated using the symbolic types "yes" and "no" to measure similarity. "Rainfall," "distance to town," and "population of town" are expressed using precise numerical values. "Flood control water level" is represented using fuzzy linguistic type, while "response level" is represented using ordinal enumeration type. The similarity calculation methods for each attribute are as follows:

(1) In cases where the attribute of the problem is symbolic, the symbolic attribute is typically represented by explicit symbols for each value. This representation provides a deterministic characterization using symbols. When the attribute values are the same, the similarity between attributes is defined as 1. If the values are different, the similarity is defined as 0. The calculation formula is as follows:

$$sim(X_j, Y_{ij}) = \begin{cases} 1, x_j = y_{ij} \\ 0, x_j \neq y_{ij} \end{cases} \qquad (6)$$

In the equation: $X_j$ represents the $j$-th attribute of the target case $X$, $Y_{ij}$ represents the $j$-th attribute of the source case $Y_i$. $sim(X_j, Y_{ij})$ represents the similarity in the $j$-th attribute between the target case $X$ and the source case $Y_i$. $x_j$ and $y_{ij}$ represent the symbol attribute values corresponding to attribute $j$ of target case $X$ and source case $Y_i$, respectively.

(2) When the attribute of the problem is a precise numerical type, the spatial distance between numbers is often used to indicate the similarity of certain numerical attributes. The similarity between attributes is calculated using the Hamming distance formula, which can be

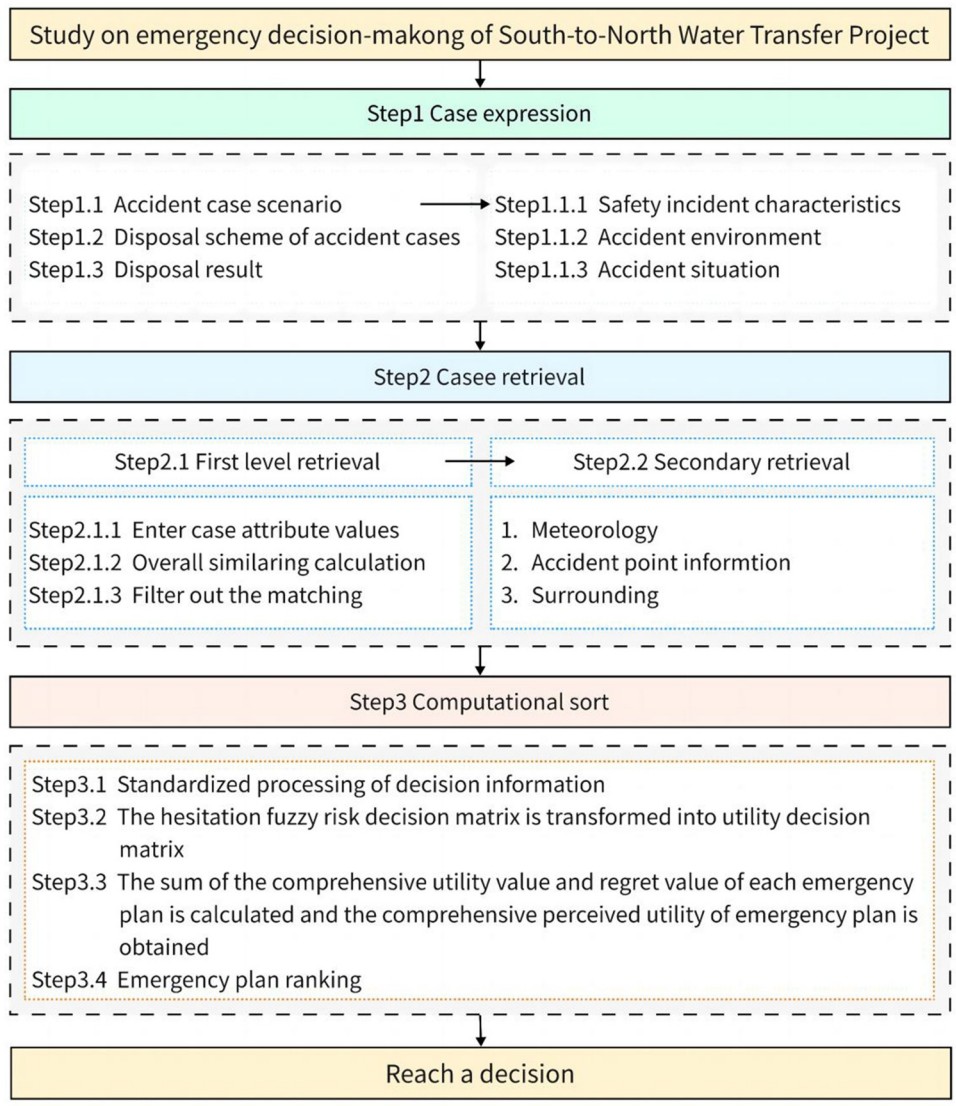

**Fig 2. Flowchart of emergency decision-making method.**

expressed as:

$$sim(X_j, Y_{ij}) = 1 - dist(X_j, Y_{ij}) = 1 - \frac{|x_j - y_{ij}|}{|\max(j) - \min(j)|} \tag{7}$$

In the formula: $x_j$ and $y_{ij}$ are the data corresponding to the $j$-th attribute of the target case $X$ and the source case $Y_i$, respectively. $\min(j)$ and $\max(j)$ represent the minimum and maximum values of attribute $j$ in the case base.

(3) When the problem attribute is a fuzzy linguistic variable, it is first converted into a triangular fuzzy number, and then the similarity of its attributes is determined. The specific calculation formula is:

$$sim(\tilde{x}, \tilde{y}) = 1 - dist(\tilde{x}, \tilde{y}) = 1 - \sqrt{(l_i - l_i^0)^2 + \frac{1}{9}[(m_i - m_i^0)^2 + (r_i - r_i^0)^2 - (m_i - m_i^0)(r_i - r_i^0)]} \tag{8}$$

In the equation: $\tilde{x}, \tilde{y}$ represents a triangular fuzzy number, corresponding to $\tilde{x} = (l_i, m_i, r_i)$ and $\tilde{y} = (l_i^0, m_i^0, r_i^0)$ respectively.

(4) When the problem attribute is of the ordered enumeration type, this type of attribute takes values in a data set with a certain level relationship, and its similarity calculation formula is:

$$sim(X_j, Y_{ij}) = 1 - |\frac{x_j - y_{ij}}{g}| \tag{9}$$

In the formula: $g$ is the number of levels for attribute $j$.

(5) The overall similarity between the source case and the target case can be obtained by calculating the similarity of various case attributes and combining them with the weights of each case attribute indicator. The specific formula for calculation is:

$$SIM(X, Y) = w_j \times sim(x_{ij}, y_{ij}) \tag{10}$$

$SIM(X,Y)$ represents the overall similarity between the target case and the source case. By considering the overall similarity, we can identify source cases that are relatively similar. The incident handling method of the selected source case can be used as the disposal plan for the current incident, providing a reliable reference for the current incident handling.

**Stage 2:** Building on the first stage's disposal plan, utilize current valid information to predict and analyze the accident's development trend, and provide corresponding disposal plans. This involves normalizing hesitant fuzzy numbers and using hesitant fuzzy utility functions to transform the hesitant fuzzy risk decision matrix into a comprehensive utility decision matrix. Furthermore, based on regret theory, calculate the regret values for implementing different emergency plans in each scenario following the occurrence of inter-basin water transfer project accidents. Then, calculate the sum of the comprehensive utility value and regret value for each emergency plan to determine the emergency plan's comprehensive perceived utility. Finally, based on the magnitude of the comprehensive perceived utility of the emergency plan, establish the emergency plan ranking.

(1) Calculation of the combined utility value

I. In order to eliminate the impact of different dimensions on the emergency response decision-making results of water diversion projects, a normalized risk decision matrix $M = [\tilde{h}_{ij}^t]_{m \times n}$ ($i \in M, j \in N, t \in L$) is constructed.

For the cost-type attribute $\tilde{h}_{ij}^t$, the normalized attribute $\tilde{h}_{ij}^t$ is:

$$\tilde{h}_{ij}^t = H(1 - \gamma_{ij}^{1(t)}, 1 - \gamma_{ij}^{2(t)}, \cdots, 1 - \gamma_{ij}^{l_{ij}(t)}), i \in M, j \in N, t \in L \tag{11}$$

For the benefit-type attribute $\tilde{h}_{ij}^t$, no normalization is required, and the normalized attribute $\tilde{h}_{ij}^t$ is:

$$\tilde{h}_{ij}^t = \tilde{h}_{ij}^t, i \in M, j \in N, t \in L \tag{12}$$

II. To compute the utility value $u_{ij}^t$ of implementing emergency plan $A_i$ in scenario $S_t$ with respect to attribute $D_j$:

$$u_{ij}^t = \left(\frac{s(\tilde{h}_{ij}^t)}{1 + v(\tilde{h}_{ij}^t)}\right)^\alpha, i \in M, j \in N, t \in L \tag{13}$$

In the equation, parameter $\alpha(0<\alpha\leq1)$ is a constant that the decision-maker assigns based on actual requirements, while $s(\tilde{h}_{ij}^{t})$ and $v(\tilde{h}_{ij}^{t})$ represent the scoring function and deviation function of the normalized attribute $\tilde{h}_{ij}^{t}$, respectively.

III. Combining attribute weights $w_j$ and utility values $u_{ij}^{t}$, the comprehensive utility value $U_{ij}^{t}$ of emergency plan $A_i$ implemented in scenario $S_t$ for attribute $D_j$ is calculated as follows:

$$U_{ij}^{t} = w_j * u_{ij}^{t}, i \in M, j \in N, t \in L \tag{14}$$

(2) Regret value calculation

Based on regret theory, in the decision-making process, decision makers typically compare the selected option with other alternative options. If choosing another option results in greater utility, the decision maker experiences regret. If choosing another option would lead to unfavorable outcomes, the decision maker experiences delight.

As decision-makers are typically risk-averse, the regret-joy function $R(\Delta V)$ is a monotonically increasing concave function, as illustrated in Fig 3, and can be represented as:

$$R(\Delta V) = 1 - \exp(-\delta\Delta V) \tag{15}$$

In the equation, $\delta$ is the regret aversion coefficient, $\delta>0$. The greater $\delta$ is, the higher the degree of regret aversion for the decision-maker. $\Delta V$ represents the utility difference of the solution.

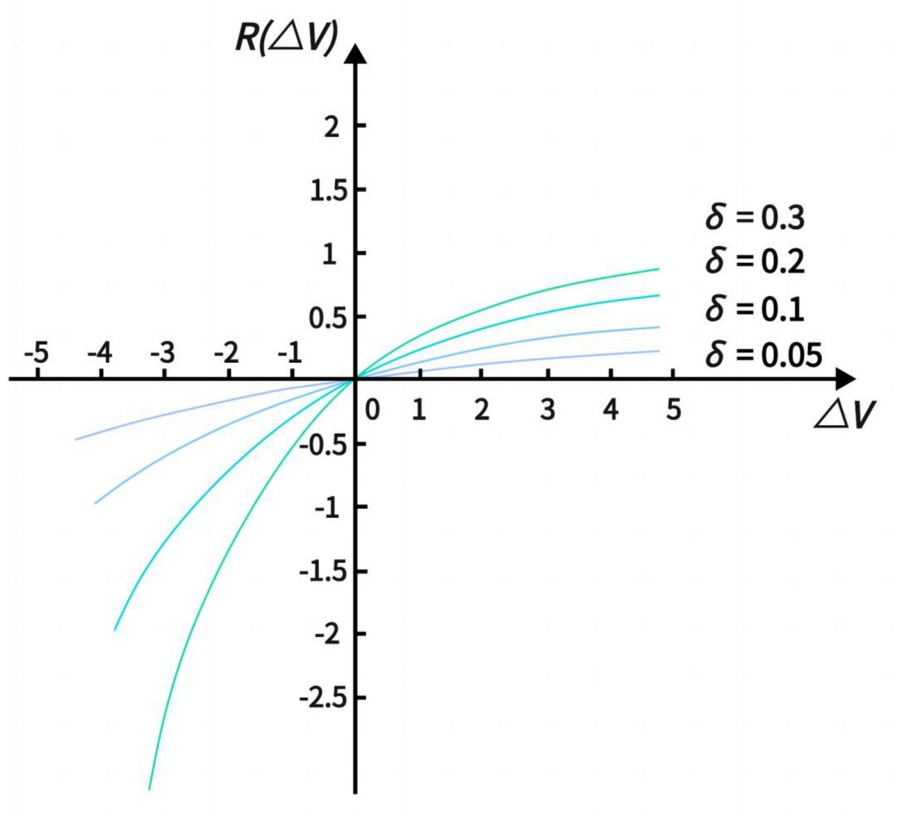

**Fig 3. Image of the regret-exultation function $R(\Delta V)$.**

Considering the equation, the regret value $R_{ij}^t$ for attribute $D_j$ after the implementation of emergency plan $A_i$ in scenario $S_t$ is computed as follows:

$$R_{ij}^t = 1 - \exp(-\delta(U_{ij}^t - U_{ij}^{t*})), i \in M, j \in N, t \in L \qquad (16)$$

In the equation: $U_{ij}^{t*} = \max\{U_{ij}^t | i = 1, 2, \cdots, m\}$, $U_{ij}^{t*}$ represent the comprehensive utility value of the best emergency plan implementation for attribute $D_j$ under scenario $S_t$.

(3) Sequencing of emergency programmes

Considering the various attributes after a coal mine accident, the perceived utility $\varphi_{ij}^t$ for attribute $D_j$ after implementing emergency plan $A_i$ in scenario $S_t$ is calculated, then:

$$\varphi_{ij}^t = U_{ij}^t + R_{ij}^t, i \in M, j \in N, t \in L \qquad (17)$$

Furthermore, by taking into account the diverse scenarios and attributes that may arise after a coal mine accident, the comprehensive perceived utility $\varphi(A_i)$ after implementing emergency response plan $A_i$ is calculated, then:

$$\varphi(A_i) = \sum_{t=1}^{l} \sum_{j=1}^{n} p_t \varphi_{ij}^t, i \in M \qquad (18)$$

Finally, emergency plans are ranked based on the magnitude of the comprehensive perceived utility $\varphi(A_i)$, where a larger $\varphi(A_i)$value indicates a better outcome after implementing emergency plan $A_i$.

## Case study

In this chapter, we use the decision-making process for a sudden backflow incident in a cross-basin water transfer project in the South-to-North Water Diversion as the research background to validate the applicability of the research methodology proposed in this paper. The current accident situation involves flooding caused by a combination of high upstream discharge and rainfall in the downstream section of the river. The water level near the intake of the backflow phenomenon is close to the design water level, and the upstream head of the intake is eroded by the flood. The emergency management department has urgently initiated a level III emergency response. To effectively respond to potential sudden incidents in the cross-basin water transfer project, the water transfer project management department has corresponding disposal plans and accident cases for reference. However, as these cases are derived from the information since the operation of the South-to-North Water Diversion Project on the one hand, and on the other hand, it needs to be combined with practical experience and consultation with the relevant experts. Therefore, this paper sets target case and 10 inverted siphon project contingency scenarios as source cases to verify the effectiveness and applicability of this paper's model based on practical experience and consultation with relevant experts. The specific data of the source and target cases are shown in Table 1.

### Stage 1

Using Formula (10), the similarity between the current flood season siphonage engineering accidents and and the source cases set out in this paper, and the results are shown in Table 2:

Through analysis of the computational results, it is evident that the similarity score $SIM(Y_8, X_0) = 0.911$ is the highest, indicating that the source case $Y_8$ is most similar to the target case $X_0$. Analyzing the specific information contained in the current accident situation, it is apparent that the historical accident details of source case $Y_8$ closely align with it. Hence, its disposal

**Table 1. Attribute values for target and source cases.**

| Case | Precipitation | Upstream flooding | Anti-flood level | Distance from town | Town population | Response Level |
|---|---|---|---|---|---|---|
| Source case $Y_1$ | 110mm/12h | No | Alert level | 1800m | 1561persons | Level III |
| Source case $Y_2$ | 90mm/12h | No | Water level | 3100m | 3562persons | Level IV |
| Source case $Y_3$ | 100mm/12h | Yes | Water level | 1300m | 375persons | Level III |
| Source case $Y_4$ | 140mm/12h | Yes | Alert level | 500m | 1875persons | Level II |
| Source case $Y_5$ | 100mm/12h | No | Water level | 2230m | 1669persons | Level IV |
| Source case $Y_6$ | 150mm/12h | Yes | Assurance level | 300m | 2571persons | Level I |
| Source case $Y_7$ | 120mm/12h | No | Alert level | 2000m | 1985persons | Level III |
| Source case $Y_8$ | 130mm/12h | No | Alert level | 1000m | 950persons | Level III |
| Source case $Y_9$ | 80mm/12h | No | Water level | 3000m | 682persons | Level IV |
| Source case $Y_{10}$ | 160mm/12h | Yes | Assurance level | 1500m | 3561persons | Level II |
| Target case $X_0$ | 120mm/12h | No | Alert level | 800m | 2355persons | Level III |

measures can be utilized to address the current accident. Choosing source case $Y_8$ as the disposal plan for the current accident is scientifically justified.

## Stage 2

Step 1: following the completion of the preliminary disposal, experts in water diversion project accident management, on-site management leaders, and rescue command leaders analyze the development trends and disposal measures for the back-siphonage engineering accident during the current flood season based on a comprehensive analysis of the on-site accident scenario. Through extensive deliberation, the expert team predicts the potential occurrence of the following three scenarios:

I. Scenario $S_1$: As analyzed by the meteorological department, the rainfall is expected to weaken, the water level of the flood is starting to decrease, the riverbank slopes are experiencing minor erosion, and the flood is beginning to scour the right bank embankment. The probability of occurrence for this scenario is 0.4.

II. Scenario $S_2$: As analyzed by the meteorological department, the rainfall is projected to continue, the flood control water level will be maintained at a constant level, the riverbank slopes will undergo continuous erosion, and hindrance to flood flow will occur at the slope section of the connecting channel of the upstream and downstream of the inverted siphon

**Table 2. Attribute values for target and source cases.**

| Case | Calculated results |
|---|---|
| Source case $Y_1$ | $SIM(X_0, Y_1) = 0.816$ |
| Source case $Y_2$ | $SIM(X_0, Y_2) = 0.648$ |
| Source case $Y_3$ | $SIM(X_0, Y_3) = 0.650$ |
| Source case $Y_4$ | $SIM(X_0, Y_4) = 0.739$ |
| Source case $Y_5$ | $SIM(X_0, Y_5) = 0.709$ |
| Source case $Y_6$ | $SIM(X_0, Y_6) = 0.653$ |
| Source case $Y_7$ | $SIM(X_0, Y_7) = 0.839$ |
| Source case $Y_8$ | $SIM(X_0, Y_8) = 0.911$ |
| Source case $Y_9$ | $SIM(X_0, Y_9) = 0.616$ |
| Source case $Y_{10}$ | $SIM(X_0, Y_{10}) = 0.538$ |

on the right bank. The safety of the masonry head of the inverted siphon on the right bank is under threat. The probability of occurrence for this scenario is 0.4.

III. Scenario $S_3$: As analyzed by the meteorological department, the rainfall is expected to intensify continuously. Upstream reservoirs have initiated flood discharge, resulting in a rapid surge of river water levels. The severe erosion of riverbank slopes poses a threat to the base of the slope. Moreover, the pipeline section exhibits hidden risks, and there is an escalated risk of flood overflow at this time. It is highly likely to breach the flood protection embankment, leading to infrastructure destruction and detrimental impacts on the safety of residents' lives and properties. The probability of this scenario occurring is 0.2.

In addition, after adopting the disposal plan of source case $Y_8$ as the initial disposal measure for the current accident, adjustments need to be made according to the development trend of the accident. There are three possible options to choose from:

I. Plan $A_1$: The plan involves setting up on-site alerts and using multimedia to remind the surrounding population to minimize outdoor activities and stay away from hazardous river areas while organizing mass evacuations. It also includes opening up rescue channels, ensuring unobstructed access for on-site rescue roads, maintaining normal electricity supply, guaranteeing the proper functioning of emergency communication equipment, organizing the orderly entry of various disaster relief equipment, rescue materials, and rescue personnel, and maintaining on-site order.

II. Plan $A_2$: Building on the implementation of the Plan $A_1$, the rescue team sets up lighting at the scene, organizes the lighting, removes the mesh fence around the headgear, and the rescue personnel produce and throw willow stones to reinforce the protective slope. They also produce and throw wire mesh stone cages to the collapsed section of the headgear, and use mechanical equipment to reinforce the backflow phenomenon of the water transfer project.

III. Plan $A_3$: Building on the implementation of Plan $A_2$, the specialized rescue team seals the breach in the flood embankment, provides protection for the eroded landslide channel slopes, raises the flood embankment, and constructs a new modular flood sub-embankment for protection.

When decision experts consider the selection of accident disposal plans, the primary attributes to be considered include three indicators: "casualties $D_1$", "economic losses $D_2$", and "social impact $D_3$" within 24 hours after the accident. Here, indicators $D_1$ and $D_2$ are cost-based indicators, while $D_3$ is a benefit-based indicator. The attribute weight vector provided by the decision experts is denoted as $w = (0.7, 0.2, 0.1)^T$.

Following this, decision experts evaluated emergency plan $\{A_1, A_2, A_3\}$ based on attribute indicator $\{D_1, D_2, D_3\}$ under three accident scenarios $\{S_1, S_2, S_3\}$. In the event of a flood season siphonage engineering accident, uncertainties arising from factors such as geological conditions and construction quality, as well as differences in opinions among decision experts, were taken into account. Therefore, evaluation information was represented in the form of hesitant fuzzy numbers, and a hesitant fuzzy risk matrix $H = [\tilde{h}_{ij}^t]_{3\times3}(t = 1, 2, 3)$ was constructed as shown in Table 3. For instance, $\tilde{h}_{12}^1 = \{0.3, 0.4\}$ indicates that when accident scenario $S_2$ occurs, emergency plan $A_1$ fulfills attribute $D_2$ to a degree of 0.3 and 0.4. Other reputation operations are outlined as follows:

I. In accordance with Eq (12), cost-type attributes $D_1$ and $D_2$ are converted to benefit-type attributes, constructing a normalized hesitant fuzzy risk decision matrix $M = [\tilde{h}_{ij}^t]_{3\times3}(t = 1, 2, 3)$ as shown in Table 4.

**Table 3. Indecision fuzzy risk matrix.**

| $A$ | $S_1(p_1) = 0.4$ | | | $S_2(p_2) = 0.4$ | | | $S_3(p_3) = 0.2$ | | |
|---|---|---|---|---|---|---|---|---|---|
| | $D_1$ | $D_2$ | $D_3$ | $D_1$ | $D_2$ | $D_3$ | $D_1$ | $D_2$ | $D_3$ |
| $A_1$ | (0.1,0.2) | (0.3,0.4) | (0.4,0.5,0.7) | (0.2,0.4) | (0.5,0.7) | (0.2,0.5,0.7) | (0.5,0.6) | (0.5,0.7) | (0.1,0.2,0.3) |
| $A_2$ | (0.3) | (0.2) | (0.2,0.5,0.7) | (0.4) | (0.5) | (0.2,0.4,0.5) | (0.8) | (0.7) | (0.2,0.4,0.5) |
| $A_3$ | (0.4,0.5) | (0.5,0.7) | (0.4,0.6) | (0.4,0.6) | (0.6,0.7) | (0.4,0.5,0.6) | (0.7,0.9) | (0.7,0.8) | (0.2,0.3,0.4) |

**Table 4. Normalised hesitant fuzzy risk decision matrix.**

| $A$ | $S_1(p_1) = 0.4$ | | | $S_2(p_2) = 0.4$ | | | $S_3(p_3) = 0.2$ | | |
|---|---|---|---|---|---|---|---|---|---|
| | $D_1$ | $D_2$ | $D_3$ | $D_1$ | $D_2$ | $D_3$ | $D_1$ | $D_2$ | $D_3$ |
| $A_1$ | (0.8,0.9) | (0.6,0.7) | (0.4,0.5,0.7) | (0.6,0.8) | (0.3,0.5) | (0.2,0.5,0.7) | (0.4,0.5) | (0.3,0.5) | (0.1,0.2,0.3) |
| $A_2$ | (0.7) | (0.8) | (0.2,0.5,0.7) | (0.6) | (0.5) | (0.2,0.4,0.5) | (0.2) | (0.3) | (0.2,0.4,0.5) |
| $A_3$ | (0.5,0.6) | (0.3,0.5) | (0.4,0.6) | (0.4,0.6) | (0.3,0.4) | (0.4,0.5,0.6) | (0.1,0.3) | (0.2,0.3) | (0.2,0.3,0.4) |

**Table 5. Combined utility matrix.**

| $A$ | $S_1(p_1) = 0.4$ | | | $S_2(p_2) = 0.4$ | | | $S_3(p_3) = 0.2$ | | |
|---|---|---|---|---|---|---|---|---|---|
| | $D_1$ | $D_2$ | $D_3$ | $D_1$ | $D_2$ | $D_3$ | $D_1$ | $D_2$ | $D_3$ |
| $A_1$ | 0.591 | 0.136 | 0.056 | 0.488 | 0.089 | 0.048 | 0.355 | 0.083 | 0.026 |
| $A_2$ | 0.526 | 0.167 | 0.048 | 0.465 | 0.115 | 0.041 | 0.193 | 0.076 | 0.041 |
| $A_3$ | 0.417 | 0.089 | 0.053 | 0.373 | 0.083 | 0.055 | 0.179 | 0.063 | 0.036 |

**Table 6. Matrix of regret values.**

| $A$ | $S_1(p_1) = 0.4$ | | | $S_2(p_2) = 0.4$ | | | $S_3(p_3) = 0.2$ | | |
|---|---|---|---|---|---|---|---|---|---|
| | $D_1$ | $D_2$ | $D_3$ | $D_1$ | $D_2$ | $D_3$ | $D_1$ | $D_2$ | $D_3$ |
| $A_1$ | 0.000 | -0.009 | 0.000 | 0.000 | -0.008 | -0.002 | 0.000 | 0.000 | -0.004 |
| $A_2$ | -0.020 | 0.000 | -0.002 | -0.007 | 0.000 | -0.004 | -0.050 | -0.002 | 0.000 |
| $A_3$ | -0.053 | -0.024 | -0.001 | -0.035 | -0.010 | 0.000 | -0.054 | -0.006 | -0.001 |

II.  Based on Formula (13), the utility value $u_{ij}^t$ of implementing emergency response plan $S_t$ for attribute $D_j$ in scenario $A_i$ is calculated. Further, the attribute weights $w_j$ and utility values $u_{ij}^t$ are pooled based on Formula (14) to calculate the combined utility value $U_{ij}^t$ of implementing contingency scenario $A_i$ against attribute $D_j$ under scenario $S_t$. A combined utility matrix is established as shown in Table 5, with the parameter set to $\alpha = 0.8$.

III.  Using Eq (16), the regret value $R_{ij}^t$ for attribute $D_j$ under emergency disposal plan $A_i$ implemented in scenario $S_t$ is calculated, and a regret value matrix is established as shown in Table 6, where the parameter is set to $\delta = 0.3$.

IV.  According to Eq (17), calculate the perceived utility $\varphi_{ij}^t$ of implementing emergency response plan $A_i$ for attribute $D_j$ in scenario $S_t$. Then, based on Eq (18), compute the overall perceived utility as $\varphi(A_1) = 0.647$, $\varphi(A_2) = 0.584$, $\varphi(A_3) = 0.422$ after the implementation of emergency plan $A$.

Based on the principle that the greater the comprehensive perceptual utility of the emergency response plan, the better the effectiveness of its implementation, the emergency

response plans are ranked as $A_1 > A_2 > A_3$. Therefore, the implementation effect of emergency response plan $A_1$ should be the best. Consequently, after a back-siphonage engineering accident occurs during the flood season, first, measures should be taken to lead the public to a safe zone and conduct basic treatment of the accident to minimize the damage of floods to riverbank slopes and the back-siphonage engineering. Second, professional emergency rescue teams should be called upon to carry out targeted and specialized salvage operations on the key points of the accident using professional equipment to ensure safety and reduce losses. Finally, reinforcements should be made to flood control embankments to effectively reduce the risk of flood overflow and river channel collapse, thereby maximizing the protection of people's lives and property.

## Discussion

To validate the differences between the regret theory and traditional objective methods, this study adopts two decision-making methods, TOPSIS and fuzzy comprehensive evaluation. Upon completing the initial case selection, the solutions are further optimized, and compared with the regret theory proposed in this study. Firstly, we need to preprocess the interval numbers of the hesitant fuzzy risk matrix in Table 2 and defuzzify it into a fixed value. Then, the matrix is standardized and substituted into the formulas of the two methods.

The TOPSIS (Technique for the Order Preference by Similarity to an Ideal Solution) method, also known as "approximation to ideal solution sorting method", is an evaluation method that ranks multiple evaluation objects as a whole based on their relative closeness to the ideal solution, in order to determine their relative superiority or inferiority. It has the advantages of flexible and convenient calculation process, as well as accurate and reasonable evaluation results [33]. Using this method involves several steps: Firstly, applying Formula (19) to the standardized emergency decision matrix to obtain the positive and negative ideal solutions $V_j^*$, $V_j^-$. Secondly, calculating the distances $S_i^*$, $S_i^-$ between scheme $D_i$ and each ideal point using Formula (20). Thirdly, determining the relative closeness degree $C_i^*$ ($0 \leq C_i^* \leq 1$) between scheme $D_i$ and the positive and negative ideal solutions based on Formula (21), where $C_i^* = 1$ and $C_i^* = 0$ represent the positive and negative ideal points, respectively. A higher relative closeness degree indicates a better scheme. The specific formulas are as follows:

$$\begin{cases} V_j^* = \max\{v_{1j}, v_{2j}, \cdots v_{mj}\} \\ V_j^- = \min\{v_{1j}, v_{2j}, \cdots v_{mj}\} \end{cases} \tag{19}$$

$$\begin{cases} S_i^* = \sqrt{\sum_{j=1}^{n}(v_{ij} - v_j^*)^2} \\ \\ S_i^- = \sqrt{\sum_{j=1}^{n}(v_{ij} - v_j^-)^2} \end{cases} \tag{20}$$

$$C_i^* = \frac{S_i^-}{S_i^- + S_i^+} \tag{21}$$

The fuzzy comprehensive evaluation method is a comprehensive evaluation and decision-making method based on fuzzy mathematics. It employs mathematical language to describe real-life problems with fuzzy boundaries and multiple levels, and provides a scientific solution using mathematical methods. In this study, $M(\wedge,\vee)$, $M(\bullet,+)$, $M(\bullet,\vee)$, $M(\wedge,\oplus)$ four operators are selected to calculate the standardized emergency decision matrix, and the higher the

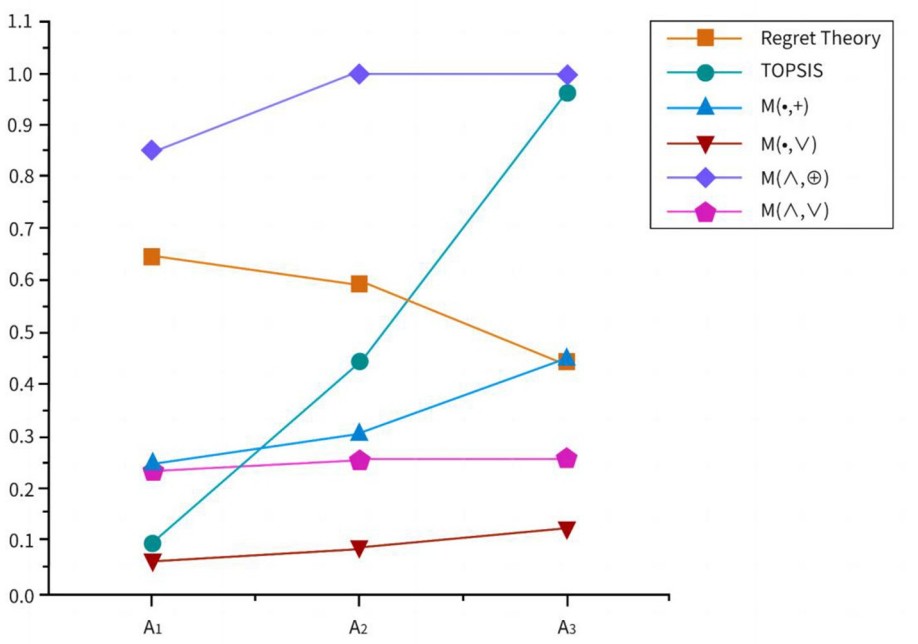

**Fig 4. Schematic comparison of results under different methods.**

comprehensive evaluation value, the better the solution. In comparison of the two algorithms, TOPSIS assumes that the relationship between evaluation indicators is linear and does not consider the mutual influence between indicators. This method is simple and easy to understand, has small computational complexity, and can intuitively reflect the degree of closeness between each solution and the ideal solution. On the other hand, fuzzy comprehensive evaluation method introduces fuzzy set theory to divide the values of evaluation indicators into fuzzy sets in order to obtain evaluation results. This method can handle the uncertainty and fuzziness between evaluation indicators, and is more suitable for decision-making problems where evaluation indicators cannot be accurately quantified in practical situations.

In Fig 4, the evaluation results of the schemes are obtained based on five mathematical calculation methods of two models. According to the table, the results obtained by the TOPSIS method and $M(\bullet,+)$, $M(\bullet,\vee)$ operator are $A_3>A_2>A_1$. The results obtained by $M(\wedge,\oplus)$, $M(\wedge,\vee)$, using two operators, are $A_3 = A_2>A_1$. The result calculated using regret theory in this paper is $A_1>A_3>A_2$, and the results obtained by multiple methods are different. Decision information is one of the important bases for emergency decision-making in unconventional emergencies. The important content of emergency decision-making is the process of information collection, processing, and feedback. Considering that the scenarios of engineering operational safety accidents are not exactly the same, different accidents have differences in weather, environment, human factors, and so on. Therefore, emergencies are characterized by sudden occurrence, high time pressure, and lack of precedent reference, which determines the extreme lack of decision information in the context of emergencies and the difficulty in obtaining it. Objective decision-making methods often fail to fully consider the actual situation and cannot achieve a correct and objective judgment based on the real circumstances. Comparing TOPSIS method and fuzzy comprehensive evaluation method, regret theory can take into account psychological behavioral characteristics of decision-makers such as reference dependence, loss aversion, diminishing sensitivity, and regret aversion. Compared to directly evaluating the

quality of solutions, regret theory can more fully consider decision-makers emotional attitudes towards different outcomes and reduce the feeling of regret caused by decision-making. In addition, regret theory can better handle uncertainties in decision-making problems, not just limited to numerical evaluation indicators. Therefore, it is of significant importance and imperative to apply regret theory in the development and further exploration of emergency plan selection methods for sudden events, taking into account decision-makers' psychological behaviors.

In conclusion, the comparative analysis results illustrate the applicability and effectiveness of the proposed regret theory approach for decision-making in engineering operational safety accident handling schemes. This suggests that the new decision-making method combining CBR and regret theory, when applied to emergency decision-making research for unexpected events in the South-to-North Water Diversion Project, is both reasonable and practical.

## Conclude

The siphon-style inter-basin water transfer project, as an important cross-basin water transfer project, its safe operation is crucial to ensure the normal water transfer of the water diversion project. From the perspective of systems engineering, this paper combines historical data and practical experience, based on CBR theory and regret theory, to calculate the perceived utility of the schemes, and then conducts a detailed analysis of the generation and dynamic adjustment of emergency plans in the accident handling of the water diversion project. The main research conclusions of this paper are as follows:

1. The selection problem of emergency plans for cross-basin water transfer project sudden events is an important issue worthy of attention. Its research can provide guidance and reference for the defense and rescue of sudden events in practical life, with practical application significance. This paper conducts decision-making research on the siphon-type sudden event in a section of the South-to-North Water Diversion cross-basin water transfer project. Based on the ranking results of the maximum comprehensive perceived utility of emergency response plans, Plan $A_1$ is finally determined as the optimal plan. In other words, when a sudden event occurs, decision-makers should first take measures to protect the public and carry out basic treatment of the accident, reducing the damage of floods to riverbank slopes and siphon-type projects. Secondly, professional emergency rescue teams should be called upon to conduct targeted and specialized rescue work on key points of the accident using professional equipment. Finally, the flood embankment should be reinforced to effectively reduce the possibility of flood overflow and river channel collapse, maximizing the protection of people's lives and property.

2. Considering the decision-maker's behavior is necessary in the selection method of emergency response plans. This paper incorporates regret theory under the consideration of decision-makers' psychological behavioral characteristics and uses TOPSIS method and fuzzy comprehensive evaluation method to optimize the plans, comparing the results with those obtained from regret theory. Finally, it is concluded that the calculation results of TOPSIS method and $M(\bullet,+)$, $M(\bullet,\vee)$ operator are $A_3 > A_2 > A_1$, the calculation results of $M(\wedge,\oplus)$, $M(\wedge,\vee)$ operator are $A_3 = A_2 > A_1$, and the results calculated using regret theory are $A_1 > A_3 > A_2$. Therefore, people often subconsciously consider regret factors before making decisions, and these psychological behaviors can have an impact on the decision analysis process and results. Considering decision-maker behavior in research is a more objective and scientific approach.

3. Emergencies possess suddenness and complexity, and existing research on decision-making methods usually aims to solve specific problems, lacking systematic and accurate

approaches. This study applies CBR theory and regret theory to investigate the emergency plan selection problem for inter-basin water transfer projects, taking the case of a siphonage emergency in a section of the South-to-North Water Diversion Project. It verifies the rationality and applicability of this new decision-making method.

The main contributions and innovations of this paper are as follows:

1. The main contribution of this study is the construction of an emergency decision-making plan for cross-basin water diversion projects, which consists of two stages. In the first stage, case-based reasoning techniques are employed to extract historical case information and disposal plans of engineering projects, and to select highly similar historical cases. In the second stage, the decision-making plans are optimized and adjusted based on the dynamic evolution characteristics of emergencies and the theory of decision-makers' regret psychology. Finally, the scientific rationality of this method is validated through practical case studies, enabling it to achieve effective multidimensional expression and rapid matching of scenarios, thereby satisfying the decision-making requirements of "scenario response".

2. Fully considering decision-makers' regret avoidance behavior, this study investigates the theoretical and practical significance of regret theory in the selection of emergency plans for emergency events. Finally, the applicability of the TOPSIS method and fuzzy comprehensive evaluation method is compared with regret theory to validate its effectiveness, which not only overcomes the impact of decision experts' bounded rationality on decision results but also provides a new direction for the knowledge of emergency plan selection methods for emergency events.

3. The accident information indicators in the inter-basin water transfer project emergency events are strictly differentiated. Considering the variety of attribute indicators in emergency events, this study describes the accident information of water transfer projects using four types: symbolic, ordinal enumeration, precise numerical, and fuzzy linguistic. Combined with the CBR method, historical case information that is more similar to the current information is selected to provide reference for the early disposal of accidents. Additionally, taking into account the excessive uncertainty factors and divergent information from decision experts after the occurrence of water transfer project accidents, hesitant fuzzy numbers are employed to represent decision information, thus enhancing the effectiveness of emergency response.

Despite this, there are still certain shortcomings in this study that need further improvement. For example: (1) How to rapidly and accurately obtain the state probabilities and attribute weights involved in emergency risk decision-making for sudden events. (2) How to fully grasp the changes in a situation in a short period of time and accurately estimate potential losses. (3) How to optimize the allocation of resources and achieve the best implementation effect in the event of conflicts arising from different sub-plans occupying the same resources in an emergency plan combination.

## Author Contributions

**Conceptualization:** Xuewan Du.

**Data curation:** Xin Huang, Xiaoxia Fei.

**Formal analysis:** Feng Li.

**Investigation:** Xin Huang.

**Methodology:** Xuewan Du.

**Resources:** Xin Huang, Xiaoxia Fei.

**Supervision:** Feng Li, Xiaoxia Fei.

**Writing – original draft:** Feng Li, Xuewan Du.

**Writing – review & editing:** Feng Li, Xuewan Du, Xin Huang.

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
