## [Decision Letter · Decision Letter 0]

26 Dec 2023

PONE-D-23-38121Study on the Methodology of Emergency Decision-Making for Water Transfer Project Contingencies: A Case-Based Reasoning and Regret Theory ApproachPLOS ONE

Dear Dr. Huang,

Thank you for submitting your manuscript to PLOS ONE. After careful consideration, we feel that it has merit but does not fully meet PLOS ONE’s publication criteria as it currently stands. Therefore, we invite you to submit a revised version of the manuscript that addresses the points raised during the review process.

please consider all comments

We look forward to receiving your revised manuscript.

Kind regards,

Ahmed Mancy Mosa, Ph.D.

Academic Editor

PLOS ONE

Journal Requirements:

4. We note that your Data Availability Statement is currently as follows: "Yes - all data are fully available without restriction"

Reviewers' comments:

Reviewer's Responses to Questions

**Comments to the Author**

1. Is the manuscript technically sound, and do the data support the conclusions?

Reviewer #1: Yes

Reviewer #2: Yes

2. Has the statistical analysis been performed appropriately and rigorously? 

Reviewer #1: Yes

Reviewer #2: Yes

3. Have the authors made all data underlying the findings in their manuscript fully available?

Reviewer #1: Yes

Reviewer #2: Yes

4. Is the manuscript presented in an intelligible fashion and written in standard English?

Reviewer #1: Yes

Reviewer #2: Yes

5. Review Comments to the Author

Reviewer #1: Comment 1: In the introduction part the objectives of importance of the study have to clearly stated

Comment 2: the overall significance of the study needs to clearly state.

Comment 3: the introduction part is very bulky. It needs to make it clear by modifying it. The research gap and objectives have to clearly provide in this part.

Comment 4: In introduction part on line 37 ‘According to statistics’ it needs to provide the reference for justification.

Comment 5: also in introduction part on line 39 ‘As a major water transfer project in China, the South-to-North Water Diversion Project has 40 transferred over 40 billion cubic meters of water’ it needs justification

Comment 6: Are you sure you are following the guideline manuscript preparation of PLOS ONE? Why include Literature review? Let check it.

Comment 7: In methodology part, line 192 where is the source of equation? Before providing the equation you have to clearly discuss how the equation used.

Comment 8: On page 235, why you use future tense ‘solution to the above problem will be addressed’’

Comment 9: on line 573 you mentioned to use both TOPSIS and fuzzy comprehensive evaluation methods it needs to clearly describe the difference between both methods and which one is more preferable for your research?

Comment 10: some references are very old and out dated (i.e. Chen, S.-M.; Tan, J.-M. Handling multicriteria fuzzy decision-making problems based on vague set theory. Fuzzy Sets and Systems 1994, 67, 163-172,doi:https://doi.org/10.1016/0165-0114(94)90084-1)

Reviewer #2: The study introduces a two-stage emergency decision-making framework tailored to the uncertain developmental trajectories of emergencies within cross-basin water transfer projects. The first stage encompasses the utilization of case-based reasoning technology to extract historical case information concerning the cross-basin water transfer project and its corresponding mitigation strategies. Employing structural similarity and local attribute similarity algorithms. Also, an overarching similarity model is constructed to identify historical cases exhibiting high degrees of resemblance. In the second stage, the work further discusses the adaptive nature of emergencies through the application of regret theory to adjust the emergency decision-making framework in response to the dynamic evolution characteristics of emergencies. Finally, the efficacy and feasibility of the proposed two-stage emergency response methodology are corroborated through empirical validation using real-world case scenarios. Thus, this manuscript is recommended for publication in this Journal. However, the following minor comments may improve the manuscript.

1. The abstract needs improvement to clarify the main topic of this paper.

2. Some keywords are missing.

3. Please revise the whole paper in terms of punctuation, especially for the equations. Some commas and full stops are missing.

4. The introduction needs to be improved and arranged properly with paragraphs to separate the previous knowledge, the research gaps, a clear input you have made and the current paper arrangement.

5. Please cite some recent relevant papers to improve the introduction part.

6. An improvement in the quality of the figures is needed.

7. Read the whole manuscript and try to remove the passive voice misuse and typo mistakes in some spots.

6. PLOS authors have the option to publish the peer review history of their article (what does this mean?). If published, this will include your full peer review and any attached files.

Reviewer #1: **Yes: **Siraj Abduro Abdulahi

Reviewer #2: **Yes: **Dr. Abdulaziz Garba Ahmad

Federal University of Technology, Babura, Nigeria

---

## [Author Response · Author response to Decision Letter 0]

7 Feb 2024

On behalf of all the contributing authors, I would like to express our sincere appreciations of your letter and reviewers’ constructive comments concerning our article entitled“Study on the Methodology of Emergency Decision-Making for Water Transfer Project Contingencies: A Case-Based Reasoning and Regret Theory Approach”(Manuscript No: PONE-S-23-50080). These comments are all valuable and helpful for improving our article. According to your comments, we have made extensive modifications to our manuscript. In this revised version, changes to our manuscript were all highlighted within the document by using yellow-colored text. The following are specific responses to each question.

Author's Reply to the Review Report (Reviewer 1)

We sincerely appreciate the valuable feedback from editor and reviewers, which we use to improve the quality of the manuscript. The reviewers' comments are presented in black, bolded font, and specific questions have been numbered. Our recoveries are given in normal font, with changes/additions to the manuscript in yellow text.

Comment 1: In the introduction part the objectives of importance of the study have to clearly stated.

Comment 2: the overall significance of the study needs to clearly state.

Comment 3: the introduction part is very bulky. It needs to make it clear by modifying it. The research gap and objectives have to clearly provide in this part.

We are sincerely grateful for your constructive comments on our manuscript. All three of these questions were asked about the introductory section of the manuscript, so we wondered if they could be answered together in combination. We have carefully revised this section in the light of your comments.

We have divided the introduction into five paragraphs. The first paragraph introduces the concepts and benefits associated with inter-basin water transfer projects and explains the background of the study. The second paragraph analyses the serious problems such as emergencies that are prone to occur in the operation of the current inter-basin water transfer project, and leads to the research purpose of this paper, which is the study of emergency decision-making and disposal of emergencies in the inter-basin water transfer project. The third paragraph explains the research methodology adopted in this paper and illustrates the current research gaps as well as the significance of this paper. The fourth paragraph briefly describes the main contributions of this study. Finally, the fifth paragraph briefly describes the arrangement of the parts of the current article. Here is the revised introduction, and we hope to get your comments on it.

Comment 4: In introduction part on line 37 ‘According to statistics’ it needs to provide the reference for justification.

We sincerely appreciate the valuable comments. We have checked the literature carefully and added more reference. 

We added the reference after the sentence ‘According to statistics, there are currently over 160 water transfer projects worldwide’, the reference is as follows: [1] Faúndez, M.; Alcayaga, H.; Walters, J.; Pizarro, A.; Soto-Alvarez, M. Sustainability of water transfer projects: A systematic review [J]. Science of The Total Environment, 2023, 860, 160500. [2] Wang, Z.; Wang, X. The Significance and Technical Key of South-to-North Water Diversion Project The Thirteenth National Academic Conference on Structural Engineering Invited Report [J]. Engineering Mechanics, 2004, 180-189.

Comment 5: also in introduction part on line 39 ‘As a major water transfer project in China, the South-to-North Water Diversion Project has transferred over 40 billion cubic meters of water’ it needs justification.

We feel great thanks for your professional review work on our manuscript. We have checked the literature carefully and added more reference. 

We changed the original sentence ‘As a major water transfer project in China, the South-to-North Water Diversion Project has transferred over 40 billion cubic meters of water since the completion of the main construction of the Middle Route Phase I and the Eastern Route, directly benefiting as many as 120 million people.’ to sentence ‘As a major water transfer project in China, the South-to-North Water Diversion Project have transferred 65.4 billion cubic meters of water since the completion of the main construction of the Middle Route Phase I and the Eastern Route, directly benefiting as many as 176 million people and replenished 10 billion cubic meters of water for ecological purposes.’ and added the argument, as shown below: [10] Mulyungi, P. South-North Water Transfer/Diversion Project in China [R]. Constructionreview, 2023.

Comment 6: Are you sure you are following the guideline manuscript preparation of PLOS ONE? Why include Literature review? Let check it.

We are sincerely grateful for your constructive comments on our manuscript. This is a big mistake of ours and we are sorry for bringing you bad feelings. We have removed the literature review from the revised manuscript and have made some changes to the formatting of the manuscript in accordance with your journal's requirements, and again apologise for our errors. We hope we can get your forgiveness.

Comment 7: In methodology part, line 192 where is the source of equation? Before providing the equation you have to clearly discuss how the equation used.

We are sincerely grateful for your constructive comments on our manuscript. We precede the description of the problem in the section on research methodology with a brief description of the research methodology used in this paper. This makes the methodology easier to understand and gives a more coherent overall structure to the article. 

Research methodology

Case-Based Reasoning (CBR) theory is a method that solves current case problems by using the solutions to similar historical cases. It helps decision-makers make choices based on similarity and implementation efficiency. If the historical cases used for reference are less relevant to the target case, the generated solutions based on the historical cases will also be less effective. Therefore, applying solutions from similar historical cases with poor implementation effects to the target case could result in poor decision-making [30]. Through case reasoning, decision-makers reasoning speed can be improved, the efficiency of emergency rescue and disposal decision-making can be increased, and the feasibility of the final plan can be ensured. To solve the problem of emergency decision-making for sudden events in the South-to-North Water Diversion Project, this study describes and analyzes the problem using this method. In the following sections, some related background knowledge is introduced to make the method more accessible.

Problem description

In the event of an unexpected incident, decision-makers retrieve historical cases from the case repository, initially assuming that the emergency case repository for a cross-basin water transfer project is denoted as......

Comment 8: On page 235, why you use future tense ‘solution to the above problem will be addressed’ .

We feel great thanks for your professional review work on our manuscript. The point we want to make here is that in the next chapters we are going to focus on solving these problems. However, due to our really very limited level of English expression, we got the tenses wrong. After asking the professionals, we changed “The specific solution to the above problem will be addressed in the following sections” in the manuscript to “In the following chapters, this study specifically addresses the above issues”.

Comment 9: on line 573 you mentioned to use both TOPSIS and fuzzy comprehensive evaluation methods it needs to clearly describe the difference between both methods and which one is more preferable for your research?

We are sincerely grateful for your constructive comments on our manuscript. We have modified the discussion section by describing the differences and advantages between the two approaches, as well as the strengths and applicability of the regret theory we used in our study. Here is what we added to the discussion, and we hope to get your comments on it.

In comparison of the two algorithms, TOPSIS assumes that the relationship between evaluation indicators is linear and does not consider the mutual influence between indicators. This method is simple and easy to understand, has small computational complexity, and can intuitively reflect the degree of closeness between each solution and the ideal solution. On the other hand, fuzzy comprehensive evaluation method introduces fuzzy set theory to divide the values of evaluation indicators into fuzzy sets in order to obtain evaluation results. This method can handle the uncertainty and fuzziness between evaluation indicators, and is more suitable for decision-making problems where evaluation indicators cannot be accurately quantified in practical situations.

Comparing TOPSIS method and fuzzy comprehensive evaluation method, regret theory can take into account psychological behavioral characteristics of decision-makers such as reference dependence, loss aversion, diminishing sensitivity, and regret aversion. Compared to directly evaluating the quality of solutions, regret theory can more fully consider decision-makers emotional attitudes towards different outcomes and reduce the feeling of regret caused by decision-making. In addition, regret theory can better handle uncertainties in decision-making problems, not just limited to numerical evaluation indicators.

Comment 10: some references are very old and out dated (i.e. Chen, S.-M.; Tan, J.-M. Handling multicriteria fuzzy decision-making problems based on vague set theory. Fuzzy Sets and Systems 1994, 67, 163-172, doi: https://doi.org/10.1016/0165-0114(94)90084-1)

We feel great thanks for your professional review work on our manuscript.We think this is an excellent suggestion. Based on your comments, we have made corrections to make the article more coherent and persuasive by replacing the old references with relevant papers from recent years. We appreciate for your warm work earnestly and hope that the correction will meet with approval.

Author's Reply to the Review Report (Reviewer 2)

We sincerely appreciate the valuable feedback from editor and reviewers, which we use to improve the quality of the manuscript. The reviewers' comments are presented in black, bolded font, and specific questions have been numbered. Our recoveries are given in normal font, with changes/additions to the manuscript in yellow text.

Comment 1: The abstract needs improvement to clarify the main topic of this paper.

We feel great thanks for your professional review work on our manuscript. Here is our improved abstract.

Abstract: In order to tackle the global water imbalance problem, a multitude of inter-basin water transfer projects have been built worldwide in recent decades. Nevertheless, given the complexity and safety challenges associated with project operation, effective emergency decision-making is crucial for addressing unforeseen incidents. Hence, this research has developed a two-stage emergency decision-making framework to tackle the uncertainty in the development trends of emergencies in inter-basin water transfer projects. (1) The first stage mainly utilizes case-based reasoning techniques to extract historical case information and disposal plans for inter-basin water transfer projects. Subsequently, a holistic similarity model is built by employing structural similarity and local attribute similarity algorithms to identify highly similar historical cases. (2) The second stage involves the optimization and adjustment of decision-making plans based on the dynamic evolution characteristics of emergencies. It utilizes the theory of decision-makers' regret psychology and combines with practical case studies to verify the scientific rationality of the method. This enables it to achieve effective multidimensional expression and rapid matching of scenarios, satisfying the decision-making requirements of "scenario response". Finally, this study compares the results obtained from this method with those computed using the traditional TOPSIS method and fuzzy comprehensive evaluation method, further validating its feasibility and effectiveness. In practice, this method can provide effective support for decision-makers' work.

Comment 2: Some keywords are missing.

We are sincerely grateful for your constructive comments on our manuscript. Thanks for your reminder. Here are our improved keywords.

Keywords: Inter-basin water transfer projects; Emergency decision-making for emergencies; Case-based reasoning; Regret theory; TOPSIS method; Fuzzy integrated evaluation method

Comment 3: Please revise the whole paper in terms of punctuation, especially for the equations. Some commas and full stops are missing.

We are sincerely grateful for your constructive comments on our manuscript. We did our best to correct some punctuation in the manuscript, as well as a few careless errors, and would appreciate your comments.

Comment 4: The introduction needs to be improved and arranged properly with paragraphs to separate the previous knowledge, the research gaps, a clear input you have made and the current paper arrangement.

We feel great thanks for your professional review work on our manuscript. We have made a lot of changes to the introduction part. We have divided the introduction into five paragraphs. The first paragraph introduces the concepts and benefits associated with inter-basin water transfer projects and explains the background of the study. The second paragraph analyses the serious problems such as emergencies that are prone to occur in the operation of the current inter-basin water transfer project, and leads to the research purpose of this paper, which is the study of emergency decision-making and disposal of emergencies in the inter-basin water transfer project. The third paragraph explains the research methodology adopted in this paper and illustrates the current research gaps as well as the significance of this paper. The fourth paragraph briefly describes the main contributions of this study. Finally, the fifth paragraph briefly describes the arrangement of the parts of the current article. Here is the revised introduction, and we hope to get your comments on it.

Comment 5: Please cite some recent relevant papers to improve the introduction part.

We feel great thanks for your professional review work on our manuscript. We think this is an excellent suggestion. Based on your comments, we have made corrections to make the article more coherent and persuasive by including relevant papers from recent years in the introduction part.

Comment 6: An improvement in the quality of the figures is needed.

We feel great thanks for your professional review work on our manuscript. We have modified Figure 2., Figure 3. and Figure 4. as much as possible in the newly submitted manuscript. I wonder if we can get your satisfaction.

Comment 7: Read the whole manuscript and try to remove the passive voice misuse and typo mistakes in some spots.

We feel great thanks for your professional review work on our manuscript. We feel sorry for our carelessness and have tried our best to polish the language in the revised manuscript. We appreciate for your warm work earnestly and hope that the correction will meet with approval.

---

## [Decision Letter · Decision Letter 1]

26 Feb 2024

Study on the Methodology of Emergency Decision-Making for Water Transfer Project Contingencies: A Case-Based Reasoning and Regret Theory Approach

PONE-D-23-38121R1

Dear Dr. Huang,

We’re pleased to inform you that your manuscript has been judged scientifically suitable for publication and will be formally accepted for publication once it meets all outstanding technical requirements.

Kind regards,

Ahmed Mancy Mosa, Ph.D.

Academic Editor

PLOS ONE

Additional Editor Comments (optional):

Reviewers' comments:

Reviewer's Responses to Questions

**Comments to the Author**

1. If the authors have adequately addressed your comments raised in a previous round of review and you feel that this manuscript is now acceptable for publication, you may indicate that here to bypass the “Comments to the Author” section, enter your conflict of interest statement in the “Confidential to Editor” section, and submit your "Accept" recommendation.

Reviewer #1: All comments have been addressed

Reviewer #2: All comments have been addressed

2. Is the manuscript technically sound, and do the data support the conclusions?

Reviewer #1: Yes

Reviewer #2: Yes

3. Has the statistical analysis been performed appropriately and rigorously? 

Reviewer #1: Yes

Reviewer #2: Yes

4. Have the authors made all data underlying the findings in their manuscript fully available?

Reviewer #1: Yes

Reviewer #2: Yes

5. Is the manuscript presented in an intelligible fashion and written in standard English?

Reviewer #1: Yes

Reviewer #2: (No Response)

6. Review Comments to the Author

Reviewer #1: (No Response)

Reviewer #2: (No Response)

7. PLOS authors have the option to publish the peer review history of their article (what does this mean?). If published, this will include your full peer review and any attached files.

Reviewer #1: **Yes: **Siraj Abduro Abdulahi

Reviewer #2: **Yes: **Dr. Abdulaziz Garba Ahmad

---

## [Editor Report · Acceptance letter]

7 Mar 2024

PONE-D-23-38121R1 

PLOS ONE

Dear Dr. Huang, 

I'm pleased to inform you that your manuscript has been deemed suitable for publication in PLOS ONE. Congratulations! Your manuscript is now being handed over to our production team.

Kind regards, 

on behalf of

Dr. Ahmed Mancy Mosa 

Academic Editor

PLOS ONE